# Directed cortico-limbic dialogue in the human brain

Ellen van Maren [1], Camille G. Mignardot[1,2], Roland Widmer [1], Cecilia Friedrichs-Maeder [1], Juan Ansó[3], Päivi Nevalainen[1,4], Markus Fuchs[1], Claudio Pollo[5], Athina Tzovara [1,2], Timothée Proix [6], Kaspar A. Schindler [1] & Maxime O. Baud [1] ✉

How can one trace the brain's orderly directed signals amid a tangle of nerve fibers? Because direct access to actual brain signaling is rare in humans, the precise wiring diagrams for cortico-limbic communication during sleep and wake remain essentially unmapped, hampering progress in neuroscience. Now, a unique neurosurgical window on the human brain allows for electrically mapping cortical connections at the hospital, but studies so far have relied on average signals, masking the dynamic nature of signal flow across brain regions and vigilance states. To causally estimate signal flow, we repeatedly probed cortico-limbic networks with short-lived electrical pulses over days and assessed the variable fate of each transmitted signal on a single-trial basis. In the resulting openly available dataset, we characterized signaling probabilities and directionality across thousands of local and long-range cortico-limbic connections over days. Challenging established views, we found that limbic structures send twice as many signals as they receive, in both wakefulness and sleep. Our findings provide a fundamental framework for causally interpreting signal flow in the brain and formulating therapeutic strategies for brain network disorders.

Electrical mapping of the nervous system, as we conceive it, originated in the 18th century, when Galvani demonstrated that electrical stimulation of a frog's sciatic nerve invariably elicits downstream crural contraction[1]. This 'animal electricity'[1] - now known as 'excitability'[2] - is the fundamental neural responsiveness that ultimately generates human thoughts. In the brain, though, as billions of nerve fibers interconnect into a complex network[3,4], inferring actual signal flow from anatomical structure remains difficult[5,6]. This structure-function challenge and limited access to the human brain[7] hamper progress in network neuroscience and the development of targeted treatments for brain network disorders.

For example, the anatomical circuit of Papez[8] extensively interconnects neocortical (orbitofrontal and temporopolar cortex) with the limbic system (hippocampus, amygdala, parahippocampal and cingulate cortex), but how communication in this circuit may process emotions[9] and memory[10–13] remains unconfirmed. As a key advantage, invasive experiments in animals can infer signal flow causally from direct probing of cortico-limbic connections[2,14–16] and temporal relationships in neuronal firing[10–12,16]. Over three decades of electrophysiological work in rodents, a two-stage model of directed cortico-limbic dialogue[13] has emerged as one possible function of the Papez circuit: (1) During awake experiences, information preferentially flows

[1]Center for Experimental Neurology, Center for sleep-wake-epilepsy, NeuroTec, Department of Neurology, Inselspital Bern, University Hospital, University of Bern, Bern, Switzerland. [2]Institute of Computer Science, University of Bern, Bern, Switzerland. [3]ARTORG Center for Biomedical Engineering Research, University of Bern, Bern, Switzerland. [4]Epilepsia Helsinki, Full Member of ERN EpiCARE, Department of Clinical Neurophysiology, HUS Diagnostic Center, University of Helsinki and Helsinki University Hospital, Helsinki, Finland. [5]Department of Neurosurgery, Inselspital Bern, University Hospital, University of Bern, Bern, Switzerland. [6]Institute for Neuroinformatics, University of Zürich, ETH Zürich, Zürich, Switzerland. ✉e-mail: maxime.baud.neuro@gmail.com

from multi-sensory to limbic cortex for salience recognition[9], memory encoding and storage[10–13]; (2) At rest, and even during sleep, the brain remains active[17,18], but information flow reverses, putatively transferring memories formed in the limbic cortex to the neocortex[10–13].

Contemporaneous non-invasive connectivity studies seeking confirmation of this hypothesis in humans have relied on functional magnetic resonance imaging (fMRI)[19] and magneto-/electro-encephalography (M/EEG)[20], which are purely correlational (i.e., non-causal) techniques, yielding a mere approximation of large-scale network dynamics[21,22]. The few invasive human studies that have actually probed limbic and neocortical signaling across the sleep-wake cycle relied on signals averaged over time[23–27], masking momentary dynamics and directionality in time-varying cortico-limbic networks[21,22]. Thus, whether cortico-limbic networks operate with a signal flow reversal during sleep in humans is currently unknown.

Using Galvani's causal approach, we aimed at mapping the large-scale signal flow empirically in the sleeping and awake cortex, testing the hypothesis of a reversing cortico-limbic dialogue directly in humans. Essentially, the same electrical probing principle applies, in which one cortical area is shortly stimulated (intracortical), and signal flow to downstream cortical areas precisely mapped via intracranial (i) EEG recordings (so-called cortico-cortical evoked potentials, CCEPs)[23–41]. Critically, unlike foundational[28,33] and recent[14,15,40,42] works with iEEG, we here assessed the variable flow of >3 million signals across the cortex at *single-trial* level, capturing their inherent dynamics over days with millisecond and millimeter precision.

## Results

We collected and analyzed iEEG from patients with epilepsy who received brain-penetrating electrodes for the recording and stimulation of seizures, as per clinical routine (Fig. 1a)[7]. Over the few days of their hospitalization, which often leads to curative brain surgery, patients may volunteer to participate in research that involves milder, unperceived intracortical stimulations[7]. Each probing stimulation delivered in a given bipolar electrode (A in Figs. 1a, 1 ms, 0.2–12 mA) located in gray matter (neocortex or limbic structures) constitutes a short-lived causal experiment (<1 s), in which evidence for signal transmission stems from a significant cortical response in any other nearby or distant recording bipolar electrode (CCEP in B or lack thereof in C, Fig.1a, b)[31]. By probing connections repeatedly, we mapped the actual flow of individual signals across brain regions and vigilance states, here novelly quantified as signaling probabilities (Figs. 1–3)[41], directionality (Figs. 2, 3)[29], and excitability (Figs. 2–4)[14].

### Cortical parcellation

To guide subsequent analyses, we first needed a parcellation of the human cortex based on its effective connectivity. The openly available[33,36] and most complete stimulation-based 'F-tract' atlas reports the *connection incidence*, calculated as the percentage of participants among 583 with a found effective connection across pairs of 77 Destrieux cortical areas[43]. As a starting point, we used the Louvain clustering method[44] to delineate 10 brain regions encompassing adjacent cortical areas that share high-incidence connectivity ('communities', color squares in Fig. 1c). This unsupervised map revealed a large-scale modular organization[3] that only slightly departs from conventional neuroanatomy delineating gyri and sulci (Supplementary Table 1)[43]. It also highlights the dense interconnectivity among limbic structures (hippocampus, amygdala, parahippocampal and temporopolar cortex), which we nevertheless analyzed separately, given the focus of our study and their established distinct functions[8,9]. Likely reflecting a true 'neural distance' (e.g., oligo- vs. polysynaptic)[45], we defined local signaling within regions as short-latency (<65 ms, 95th percentile), whereas cortical responses from long-range (≥25 mm) signaling across regions could have short- or long-latency (≥65 ms, Fig. 1d)[33], typically with a peak latency of 65–200 ms (Fig. 1b).

### Signaling probability

Using this cortical parcellation as a grouping variable for each of our participants (Inselspital cohort, N = 15, 9 females, Fig. 1e, Supplementary Table 2), we next asked: Given an effective connection, *how often* in time does it transmit signal in one direction versus the other? Critically, and unlike other works[23–41], we assessed signal flow at the single-trial level, revealing its variable and dynamic nature. Upon repeatedly and randomly probing (median [IQR] 288 [119, 320] trials, Supplementary Fig. 1) each found effective connection (total #37,850) with a fixed single-pulse stimulation (3 mA, 1 ms) over days (46 h [36, 58], Fig. 1f, g), we observed large variations in the response magnitude (line length, Eq. 1 in Methods, Supplementary Figs. 2-4) or the entire lack thereof (Fig. 1f). This (in)consistency in signal flow was expressed as the *signaling probability* (P), calculated as the number of significant single-trial cortical responses observed in electrode B out of the total number of stimulations delivered in electrode A (Fig. 1f). For each single-trial, the significance of a cortical response was tested against the expected signal variability found in non-stimulated baseline recordings (i.e., null distribution, see Methods, Supplementary Fig. 5). We contrasted two maps: (1) the static F-tract map, here re-analyzed (Fig. 1c), establishes the *incidence* of effective connections across participants, based on average signals (~20 trials)[33]; (2) our dynamic map of *signaling probabilities* (Fig. 1h) reflects the individual rate of actual signal flow probed over hundreds of single-trials in each of the effective connections similarly found in our 15 participants (Inselspital, Supplementary Figs. 6, 7).

Thus, our single-trial framework goes beyond the assessment of effective connectivity, as it allows for capturing time-varying signal flow across effective connections. We estimated a dynamic cortical topology in which the relative proximity between data-derived brain regions corresponds to signaling latency[45] (e.g. cingulate cortex is close to orbitofrontal and amygdala) and the links to signaling probabilities (Fig. 1i). Although this topology is undoubtedly undersampled and biased – iEEG is collected according to clinical priorities[7] – it here serves as a reference 'connectogram' for mapping signal flow across brain regions and vigilance states.

### Signaling directionality

Next, we evaluated *signaling directionality* from the ratio of signaling probabilities in one ($P_{AB}$) and the other ($P_{BA}$) direction (Eq. 2 in Methods). Thus defined, our directionality index DI → 0 for bidirectional signaling ($P_{AB} \simeq P_{BA}$) and DI → ±1 for unidirectional signaling in either direction ($P_{AB} \not\simeq P_{BA}$, Fig. 2a), characterizing an effective connection's dynamics beyond response magnitudes or latencies (Supplementary Fig. 8). We found that cortical responses in 3,368 local connections within the same brain region (distance <25 mm) were invariably present (median [IQR] P = 99% [96, 100]), always short-latency (<65 ms) and bidirectional (|DI| = 0.01 [0.00, 0.14], Fig. 2b). In contrast, long-range (≥25 mm) signaling across brain regions – tested within hemisphere – tended to be more dynamic with (1) a majority (75%) of short-latency connections signaling with variable signaling probability (P = 67% [26, 96]) and directionality index (|DI| = 0.52 [0.14, 1]), and (2) rarer (25%) long-latency connections (≥65 ms) signaling with low signaling probability (P = 21% [13, 40]) and unidirectionally (|DI| = 1.00 [0.63, 1.00], p = 0 for P and DI, one-way Kruskal-Wallis, Fig. 2b). Based on this result, we speculate that long-range signaling may help broadcast neural communication at short-latency and create directional feedforward or feedback loops at long-latency.

### Connection excitability

In a complex network, *how often* a signal is transmitted along a connection (i.e., $P_{AB}$) may depend on *how excitable* (i.e., responsive) this connection is. To test this, we quantified our previously validated *excitability index* (ExI)[14,46] in a subset of connections (total #2365) as the area under the stimulation-response curve for 14 stimulation intensities

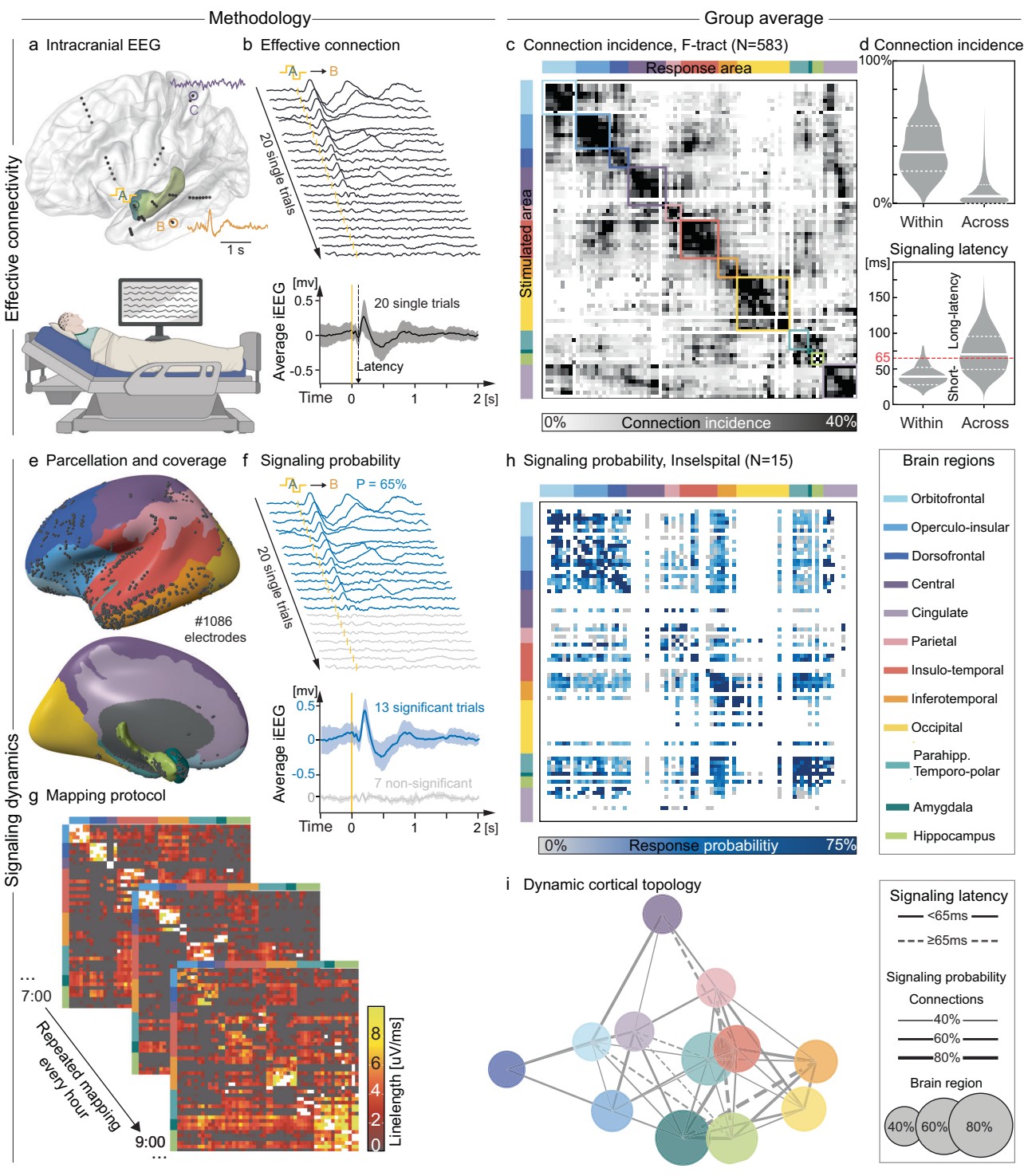

from 0.2 to 12 mA delivered randomly in triplicates over 15 min (Fig. 2c), where ExI→0 for non-excitable and ExI→1 for maximally excitable connections (Supplementary Fig. 9). On average, excitability slightly decreased ($p < 10^{-8}$, Kruskal-Wallis, Fig. 2d) from local (ExI [IQR]: 0.64 [0.56, 0.75]) to long-range short-latency (0.62 [0.54, 0.71], $p = 0.01$, post-hoc Mann-Whitney U) and long-latency connections (0.58 [0.51, 0.67], $p < 10^{-6}$). Yet, among the characteristics of individual connections (inflated Euclidean distance, signaling latency, response magnitude and ExI), signaling probabilities best correlated with the ExI (Pearson's rho = 0.60, $p < 10^{-10}$, Fig. 2e, Supplementary Fig. 9). Thus, upon probing, the invariable immediate local cortical response variably propagates to distant cortical areas, possibly via routing mechanisms dependent on a connection's excitability.

## Cortico-limbic signaling

Focusing on long-range connections, we indeed found a region-specific ($p < 0.05$) and region-pair specific ($p < 0.05$) influence on signaling probabilities during wakefulness, using a logistic regression model across participants that accounted for the observed decrease in signaling probability by distance ($p < 0.001$, pseudo $R^2 = 0.18$, Eq. 3 in Methods, Supplementary Fig. 8)[35]. All brain region-pairs within hemispheres encompassed a mix of bidirectional and unidirectional

**Fig. 1 | Effective connectivity and signaling probabilities. a** Intracortical recording of brain activity via implanted electrodes (iEEG, top) in a hospitalized participant (bottom, schematic created in BioRender. Baud, M. (2025) https://BioRender.com/bahw5m4). Dots represent iEEG bipolar electrode contacts in limbic structures (e.g., A) or neocortex (e.g., B, C, colors in legend). **b** Effective connection with 20 single-trial examples (ranked) and mean ± SD iEEG traces showing cortico-cortical evoked potentials (CCEPs) of varying magnitude in the inferior temporal gyrus (B) upon repeated bipolar electrical stimulations in the ipsilateral amygdala (A at $T_0$: 1 ms, 3 mA, yellow biphasic waveform and ticks). Dashed arrow: signaling latency. **c** Ordered adjacency matrix with parcellation of brain regions (color squares, 'Louvain communities') based on the *incidence* of effective connections between any two of 77 Destrieux cortical areas among 583 participants in the F-tract study (v2210). **d** Distribution, median (full) and IQR (dashed) of connection incidences (top) and signaling latency (bottom) within and across brain regions. Signaling latency threshold set at the 95th percentile within brain regions

(dotted red line). **e** Resulting color-coded parcellation projected onto the MNI inflated left hemisphere. Black dots: bipolar electrodes in gray matter available from the left or right hemisphere across our 15 participants. **f** Same example iEEG traces as in **b**, here with cortical responses tested statistically at the single-trial level and mean ± SD signal separately for significant (blue) and non-significant trials (gray). **g** Illustration of the stimulation protocol wherein all electrodes are probed 3-5 times each hour to repeatedly generate maps of cortical signal flow within and across brain regions (here, mean response magnitude quantified as iEEG line length). **h** Map of signaling probability averaged within a pair of cortical areas and then across 1–15 participants over hours of wakefulness. Note that the map is undersampled and shown for comparison with **c**. **i** Estimated dynamic cortical topology, in which bidirectional inter-regional signaling has different probabilities (connection thickness and size of dot) and latencies (full/dashed connection and distance between dots), that is 'neural distances'[45].

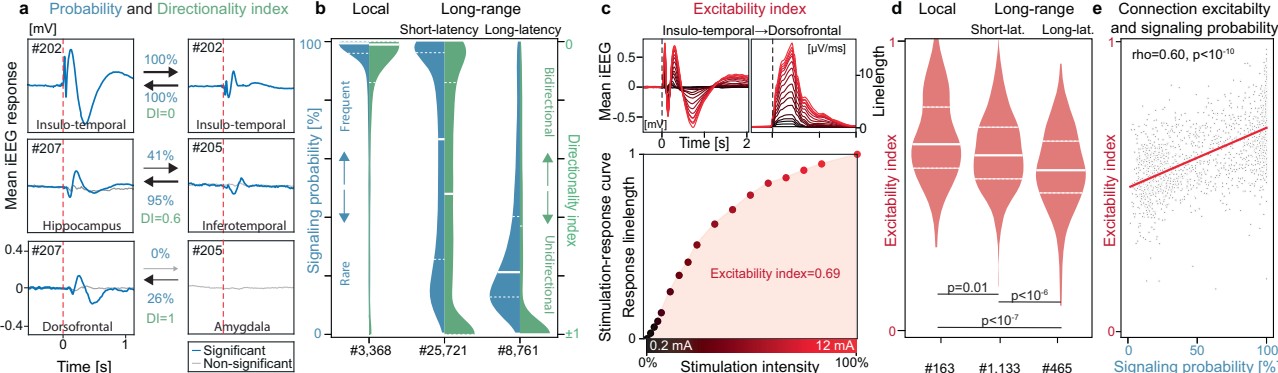

**Fig. 2 | Signaling directionality and connection excitability. a** Calculation of the directionality index (DI) based on signaling probabilities (%) in three example connections (rows) probed >200 times (#) over days in either direction (columns). For each connection, stimulation on either side is at the arrowtail and the response at the arrowhead. Average significant (blue) and non-significant (gray) trials are overlaid and plotted against the same scale. Note how signaling across connections over time can be reciprocal (top), fully asymmetric (bottom), or intermediary (middle), as quantified by the DI. **b** Distribution (probability density function against the x-axis), median (full line) and IQR (dashed) of signaling probabilities (left blue half-violin) and absolute DI (right green half-violin) in three classes of connections: local (preset <25 mm) and long-range (≥25 mm) short-latency

(<65 ms, see Fig. 1d) and long-latency (≥65 ms) connections across brain regions. **c** Calculation of the excitability index (ExI) based on examples of response magnitude growing with stimulation intensity (0.2 to 12 mA, black to red gradient) shown as the mean iEEG cortical response (top left). The corresponding linelength transform (top right) quantifies the response magnitude in a stimulation-response curve (bottom), whose area-under-the-curve is the excitability index.

**d** Distribution, median (full line) and IQR (dashed) of the ExI across a subset (# total number) of local and long-range effective connections tested with a two-sided Wilcoxon rank-sum test. **e** Two-sided Pearson correlation tested between the ExI and the signaling probability for the subset of connections in **d**.

connections, but some had biased distributions (connection-level analysis, Fig. 3a, b), resulting in consistent inter-regional directionality (participant-level analysis, Fig. 3c, d). Across participants, we found that limbic structures were mostly sending signal to all neocortical regions[34] with average P of 40–60% and DI of +0.61 (amygdala, $p < 10^{-9}$) and +0.35 (hippocampus, $p < 10^{-4}$, two-sided Wilcoxon signed-rank test, Fig. 3c). Notably, the amygdala and hippocampus signal with high probability and strong efferent directionality to the orbito-frontal, operculo-insular and cingulate regions (Fig. 3e)[8], known to be involved in the processing of memory and emotions[9,38]. This finding quantifies signaling directionality along the known anatomy of the Papez circuit and its extension to the orbitofrontal cortex by MacLean[8]. The preponderance of efferent limbic signaling could result from the observed higher excitability in limbic efferences versus afferences (Fig. 3f, Supplementary Fig. 9). In a control analysis, we found, like others[37], equivalent signaling properties among cortico-limbic connections which did or did not pertain to the epileptic network ($p > 0.05$ for both, two-way Kruskal-Wallis, Supplementary Fig. 10), indicating the likely physiological role of the limbic structures in broadcasting signal.

### Cortico-cortical signaling
By comparison with cortico-limbic connections, signaling was generally bidirectional among the nine neocortical regions (Fig. 3d,

Supplementary Fig. 11), with at most slight directional biases (|DI| < 0.2) for the orbitofrontal and inferotemporal (receiving), as well as the central and dorsofrontal regions (sending, Fig. 3). We here provide our data-derived connectogram as a graphical user interface to explore the details of the connectivity we found (see Data Availability).

### Cortico-limbic signaling during sleep
Finally, to address the long-standing hypothesis of flow reversal in cortico-limbic signaling[10–13], we probed connections repeatedly during sleep and compared our measurements to the results above obtained during wakefulness. Thirteen participants had sufficient quantity NREM sleep (median [IQR] = 21.3% [17.1, 29.8] of time recorded) and REM sleep (4.3% [3.5, 6.1], Fig. 4a, b, Supplementary Table 2) as assessed visually by scalp EEG (Supplementary Fig. 12). Within connections, signaling probabilities were highly correlated between wake and NREM sleep (Pearson's rho = 0.90) or REM sleep (rho = 0.95, Supplementary Fig. 13), indicating overall stability of our estimated brain-wide topology across states. Still, many cortico-limbic connections had subtle but significant increases in excitability (ExI) during sleep vs. wake (median ΔExI from +7 to +10% in NREM, and from +4 to +8% in REM, Fig. 4c–f). Constituting a notable exception and contrary to the prediction, hippocampal efferences to neocortex showed a general decrease in excitability during NREM (median ΔExI = −6.3%,

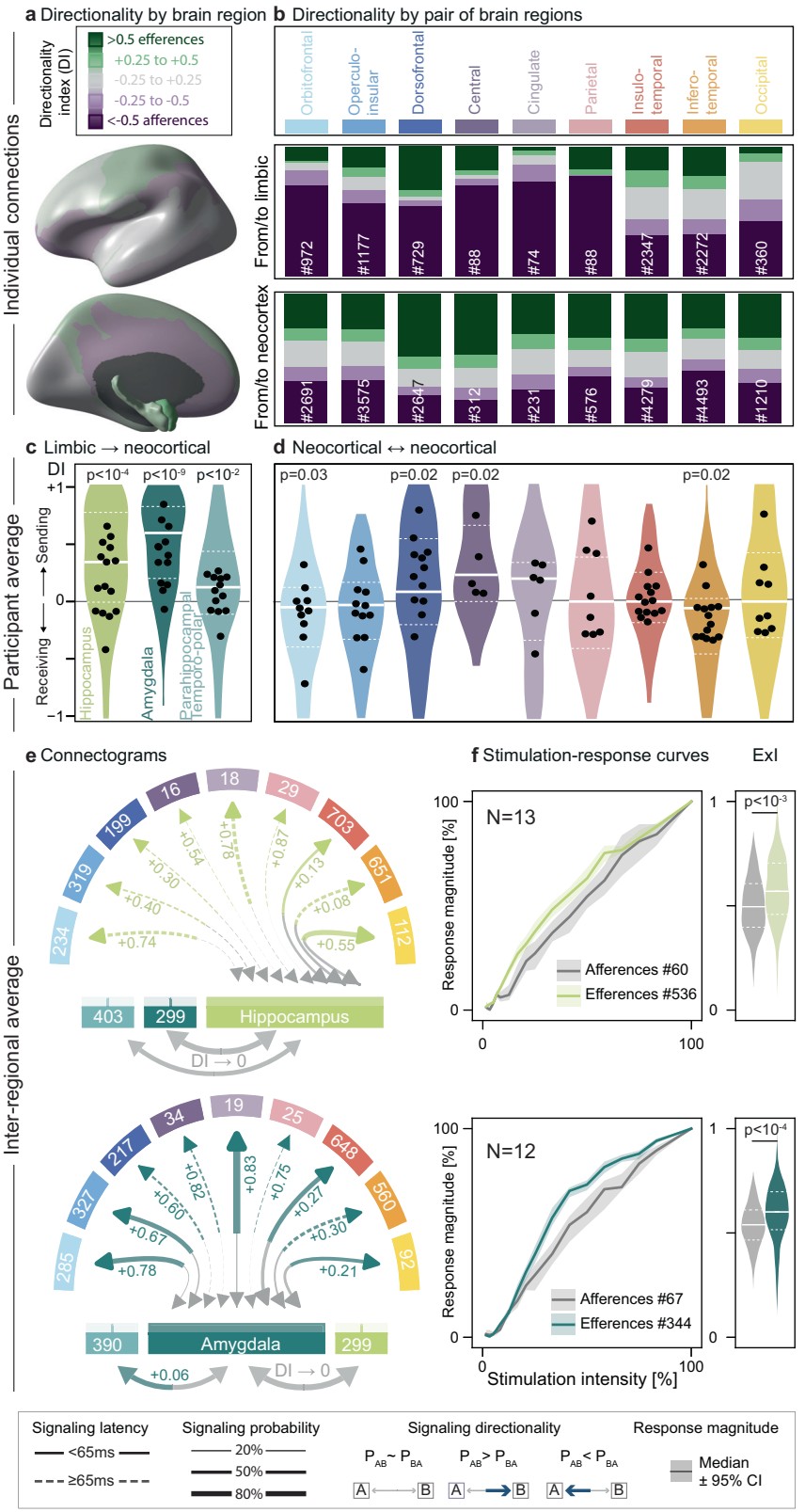

**a** Directionality by brain region **b** Directionality by pair of brain regions

**c** Limbic → neocortical **d** Neocortical ↔ neocortical

**e** Connectograms **f** Stimulation-response curves ExI

$p < 10^{-5}$, post-hoc Wilcoxon signed-rank test) and REM sleep (−5.1%, $p < 0.05$, Fig. 4f1). This was accompanied by decreased signaling probabilities and magnitude from the hippocampi ($p < 0.05$, mixed linear model, Supplementary Fig. 13) that transmitted fewer and weaker signals, notably to the cingulate cortex (-25% reduction, Supplementary Fig. 13). Thus, we did not record any flow inversion during sleep upon directly probing signal transmission to and from the hippocampus.

## Discussion

Here, we directly probed the human cortex with short-lived electrical stimulations, providing an open dataset capturing the flow of >3 million evoked signals precisely mapped over unprecedented durations. We found that, unlike a frog's nerve[1], cortico-limbic connections do not signal with one-to-one fidelity but are instead routed according to ever-changing paths[6], with regularities in wake and sleep that we uncovered. Whether artificially probed macroscopic signals reflect

**Fig. 3 | Cortico-limbic signaling. a** Mean directionality index (DI) is shown on an inflated brain for long-range connections from each brain region to all other regions, averaged within and across participants. Green regions are biased towards efferences (i.e., sending), purple towards afferences (i.e., receiving). **b** Connection-level analysis: proportion of effective efferences (green), afferences (purple), and bidirectional connections (gray) grouped by DI values (legend with cutoffs in **a**). Top: long-range cortico-limbic connections (# total number). Note the high rate of almost unidirectional connections from limbic structures to certain neocortical areas (dark purple). Bottom: long-range connections from one neocortical to any other neocortical region. For example, the dorsofrontal cortex receives many afferences from limbic structures (purple > green), and sends many efferences to other neocortical regions (green > purple). **c, d** Corresponding participant-level analysis: distribution, median (full line) and IQR (dashed) of the DI by brain region, along with averages within participants (black dots) and statistical testing with post-hoc two-sided Wilcoxon signed-rank test with FDR correction across participants. **e** Corresponding connectograms showing the inter-regional average signaling probability (line width on the receiving end) and latency (dashed vs. full lines), colored for significant directionality (DI indicated, post-hoc two-sided Wilcoxon signed-rank test with FDR correction, $p < 0.05$) from hippocampus (top) or amygdala (bottom) to neocortical brain regions (# connections averaged shown). Connectograms for pairs of neocortical regions are in Supplementary Fig. 11. A graphical user interface to visualize signaling in individual connections can be downloaded at https://github.com/neuro-elab/EvM_Connectivity/releases. **f** Average (±95% CI) stimulation-response curves (left) at stimulation intensities 0.2–12 mA in a subset of connections and corresponding ExI for hippocampal (top) and amygdala (bottom) efferences (colored) and afferences from neocortical stimulations (gray), along with post-hoc two-sided Wilcoxon rank-sum test (no FDR correction). Source data for **c**–**f** are provided in the Source Data file.

true neuronal communication is a question for future research. For now, our unique causal approach to signal flow[47] offers a radical alternative to correlational studies using passive fMRI[48] or (i)EEG[49] recordings that cannot measure directionality or excitability with certainty, even though some statistical techniques seek to do so[5,21,48]. As one step towards the holy grail of a complete dynamical connectome, our study's strength is its simplicity and directness.

Nevertheless, some limitations pertain to the collection of sparse and biased iEEG that target altered brain networks in people with focal epilepsy[7]. To avoid frequent inaccuracies in assessing global connectivity from partially sampled networks[30,31], we developed time-based metrics focused on individual connections. To increase coverage, we pooled samples from epilepsies of different laterality and focality as well as iEEG from healthy parenchyma, as usually done in the field[7,31,36]. Despite this diversity, we observed shared patterns across participants, and no significant difference between epileptic and non-epileptic cortico-limbic networks in line with others[28,37], attesting to the robustness of our findings discussed in detail below.

First, we observed that invariable local cortical excitation led to occasional signal flow to distant brain regions[39], captured in terms of signaling probabilities (P)[41]. Others have used similar intracortical stimulation methodologies, but their analyses relied on strong response magnitude in averaged signals[23–41] with a focus on signal variance[39,42], power[25] or complexity[26,42]. Beyond the assessment of effective connectivity, our single-trial framework marks a significant advancement in precisely mapping the millisecond transmission of each delivered probing signal, affording the need to capture momentary signal flow that was lost in the averages of prior approaches.

Second, our directionality index (DI) unraveled locally reciprocal connections (i.e., DI → 0), but asymmetric signal flow at long-range[39] (i.e., DI → 1). Average inter-regional directed signaling resulted from the biased assemblage of connections in either direction. Most strikingly, limbic structures broadcast[33,34,37] along the extended Papez circuit and beyond, sending about twice as many signals as they receive (i.e., DI ~ +0.5). In other words, iEEG uncovers strongly directional connections with millimeter precision in all cortices, but strong resultant biases at the scale of brain regions (i.e., centimeter) concern only the amygdala and hippocampus. Asymmetries in cortico-limbic signaling possibly shape time-varying networks[21,22] and constrain higher-order signal routing[3], notably during sleep. By comparison, neocortical signaling was largely bidirectional (i.e., DI → 0), which may enable making predictions[50], detecting errors[51], and providing feedback[52], during awake cognitive engagement.

Third, we measured the excitability index (ExI) in a subset of effective connections over minutes and found that it strongly correlated with signaling probabilities over hours, suggesting that *how often* a connection transmits signal relates to *how excitable* it is. The excitability of limbic afferences and efferences was subtly and bidirectionally modulated during sleep. Contrary to the prediction, we did not find a reversed flow from limbic to neocortical regions during sleep[10–13]. Rather, we observed a general decrease in the probability

and strength of efferent signals from the hippocampus to neocortical regions both during NREM and REM sleep, with a particularly marked reduction to frontal and cingulate cortices (−10 to −25%). Prior iEEG studies during sleep[23–27] variably found weaker[27] or stronger[23,25] average cortical responses compared to wake, likely because they did not investigate region-specific effects, which was here key to disentangle sleep modulations in opposite directions in amygdalar and hippocampal connections.

Our causal assessment of directional signal flow over days in the human cortex is unprecedented and opens new paths to uncharted territories. From a fundamental standpoint, could signal flow relate to cortical functions and cognitive performance? From a clinical standpoint, is signaling directionality altered in brain network disorders, and is it targetable by neuromodulation? More broadly, current network models seek to account for directional delays[45], which may have a large functional impact in complex networks like the brain. Our openly available dataset can help advance modeling work accounting for such dynamics. Undoubtedly, mapping cortical signaling with finer parcellation will require denser and larger datasets, as well as converging evidence from healthy humans. In the meantime, our understanding of the dynamic brain architecture is *galvanized* by volunteers among people with neurological disorders.

## Methods
### Inselspital participants
Original data analyzed in this study were collected between 2019 and 2024 from 15 adults (median [range] age 33 [19 to 63]) with pharmacoresistant epilepsy undergoing invasive presurgical evaluation at Inselspital Bern, Switzerland, who consented to participate in research. Most participants were employed (3 unemployed or working in a protected workshop). Participants had a variety of epilepsies (9 mesiotemporal, 6 other locations, Supplementary Table 2) that required a unilateral (7) or bilateral (8) implant. Sex and/or gender were not considered in the study design, nor in the inclusion criteria. Low sample size prevented sex- or gender-based analyses. Exclusion criteria were the presence of a porencephalic cavity, which would have severed connectivity. This study was approved by the Ethics Committee of the Canton Bern ID 2018-01387.

### Cortical parcellation
To guide the analysis of our own data, we first derived a cortical parcellation from the open-source F-tract dataset acquired in four French hospitals that provides the *incidence* of effective connections across 583 young patients (mean ± SD age 33.5 ± 10) with a variety of pharmacoresistant epilepsies (38% temporal, 25% frontal, rest in other locations)[33,36,53]. Mono- or biphasic stimulating pulses of 1–3 ms were delivered at 1 Hz with 1–5 mA intensity. Detection of effective connections was based on the magnitude of the average post-stimulation signal compared to the variance of the pre-stimulation baseline, with a cutoff at Z-score >5[33]. The *connection incidence* for any two of

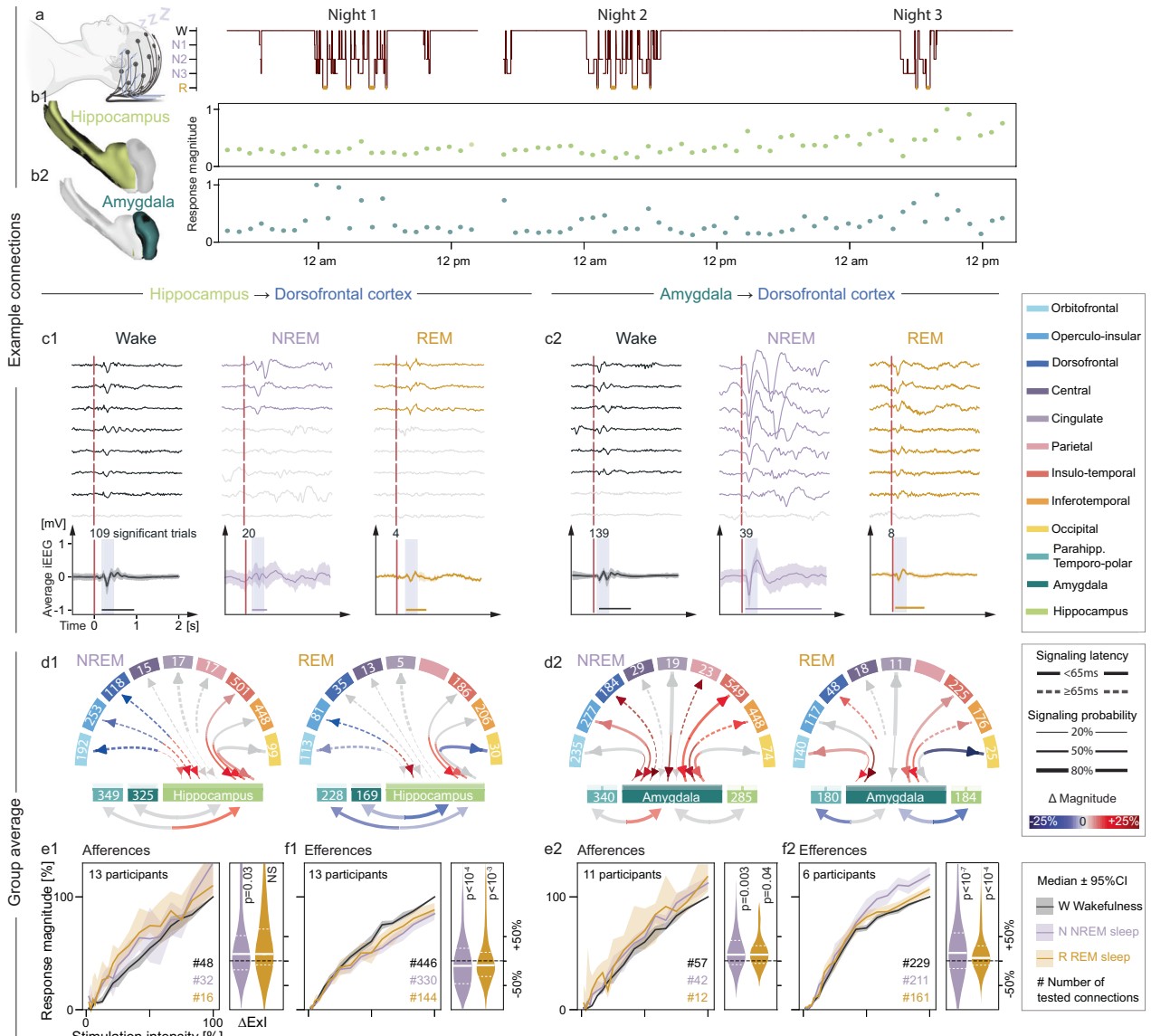

**Fig. 4 | Cortico-limbic signaling during sleep and wakefulness. a** Example of visually scored hypnogram from scalp EEG in one participant. Schematic created in BioRender. Baud, M. (2025) https://BioRender.com/0i46iia. **b** Corresponding cortical response magnitude upon stimulation in the hippocampus (light green, **b1**) or amygdala (dark green, **b2**) showing modulation by sleep. **c** Examples of eight iEEG responses in the dorsofrontal cortex upon stimulation in the hippocampus (**c1**) or the amygdala (**c2**) recorded in one electrode in one individual across vigilance states (W, N, R legend in figure bottom). Signaling probability is calculated from the significant responses (colored, non-significant are gray) out of the total number of stimulations delivered. Bottom: average (±SD, shading) of significant responses. The response magnitude is calculated in the shaded window and shown as a horizontal line to illustrate the stretched length of the response signal (line length). **d1-2** Connectograms showing the mean percentage increase (red gradient) or decrease (blue gradient) in response magnitude during sleep (NREM left, REM right) compared to wakefulness. Significant modulation for a given pair of brain regions is shown as a colored line (p < 0.05 on two-sided Wilcoxon signed-rank test with FDR correction, non-significant in gray). **e, f 1-2** Median stimulation-response curves (95% CI shaded) and corresponding ExI for varying response magnitude across vigilance states (normalized to the max response) upon stimulation at a range of stimulation intensities (0.2–12 mA, normalized from 0 to 1). Stimulations in the hippocampus or amygdala were recorded in all available neocortical regions (efferences in **f**), and stimulations in one or two neocortical regions were recorded in the amygdala and hippocampus (afferences in **e**, # connections indicated). Right: corresponding NREM and REM sleep modulation of the excitability index with p-values from the two-sided Wilcoxon signed-rank test. NS: not significant. Source data for **d**–**f** are provided in the Source Data file.

the covered 77 Destrieux cortical areas was calculated as the percentage of effective connections found across participants. We applied the Louvain clustering method[44] (resolution =2.0) to group effectively connected Destrieux cortical areas into larger, interconnected brain regions (so-called communities). We used these brain regions as a grouping variable for the rest of the study, but kept the amygdala and hippocampus separate from the surrounding parahippocampal and temporo-polar cortex, given the focus of our study and their unique functions[8].

## Electrode implantation and localization

Electrodes were implanted as clinically necessary for seizure localization and without relation to the present research study. A median [IQR] of 9 [4, 13] depth electrode leads containing 8–18 macro contacts of ⌀0.8 mm × 2 mm length spaced by 3.5 mm (Microdeep® electrodes, DIXI medical, Marchaux – Chaudefontaine, France) were implanted under general anesthesia using a stereotaxic frame. Immediately after implantation, a brain CT was obtained and aligned to the participant's presurgical MRI, using Advanced Normalization Tools[54] to localize the

electrodes in the native MRI space (Lead DBS). MRIs were spatially normalized to a standard template from the Montreal Neurological Institute (MNI) using the Statistical Parametric Mapping (SPM) algorithms. The pial surface was extracted with FreeSurfer[55]. We used the Destrieux parcellation[43] for automated electrode labeling, which was visually checked by a neurologist (MOB), as per clinical routine. Electrode contacts located at >2 mm from gray matter were excluded from further stimulation and recording. Virtual bipolar electrode coordinates (here 'electrodes') were calculated as the midpoint between neighboring physical electrode contacts. Inter-electrode 'inflated' Euclidean distances were estimated in millimeters after translocating the electrode coordinates to an inflated MNI brain.

### Recording and stimulation

Long-term intracranial and scalp EEG were recorded simultaneously and continuously over 5–14 days with a sampling frequency of 1024 Hz using the Quantum® LTM Amplifier (Natus, Middleton, Wisconsin, USA). The physical reference electrode was placed on the scalp, near the vertex. Signals from neighboring electrode contacts were subtracted to obtain bipolar recordings. Each hourly block of cortical mapping captured (1) a 5-min baseline non-stimulated period, (2) a full map based on 3 mA single-pulse stimulations repeated 3-5 times in all available bipolar electrodes in gray matter (electrode 'A') while recording from all others (electrode 'B', 'C', etc.) and (3) a stimulation-response curve from four selected stimulation electrodes (two limbic, two neocortical), relying on 14 single-pulse intensities from 0.2 to 12 mA, each repeated thrice. Single-pulse electrical stimulations (1 ms biphasic square-wave, 0.5 to 12 mA) were delivered every $4.5 \pm 0.2$ s (Gaussian-distributed jitter) in random order, using an ISIS neurostimulator coupled to a switch matrix for rapid programmatic switching between recording and stimulation electrodes (Supplementary Fig. 1a, Inomed Medizintechnik GmbH, Germany). An efferent copy was sent from the stimulator to the amplifier to mark times of stimulation ($T_0$). Hourly blocks were repeated over many hours of sleep and wake to capture signal flow (Supplementary Table 2).

### Signal preprocessing

Stimulation and switch matrix artifacts were located and removed using linear interpolation in time (10–15 ms, Supplementary Fig. 1b). Each recorded signal was then bandpass-filtered (0.5–200 Hz) and notch-filtered (50 Hz and harmonics), rescaled to a shared amplitude, and downsampled to 500 Hz (Supplementary Fig. 1c).

### Signal processing

Post-stimulation signals were evaluated for the presence of a cortical response locked to the stimulation time ($T_0$, so-called cortico-cortical evoked potentials, CCEPs). Beyond prior approaches[23–41], cortical response characteristics were assessed in single-trial (probability, magnitude, waveform) and/or averaged signals (effective connection, magnitude, latency), which required careful consideration regarding the reliability of methods to do so (Supplementary Figs. 2–5).

**Magnitude.** To calculate the magnitude of cortical responses, we favored the line length (LL) calculation[14] over other metrics given the responses' variable waveforms (Supplementary Fig. 3):

$$LL = \frac{\sum_{i=1}^{N} |x_i - x_{i-1}|}{N} * \frac{sf}{1000} \qquad (1)$$

where $N$ is the number of datapoints over which the LL is calculated, $sf$ is the sampling frequency, and $x$ is the iEEG signal. A 250 ms window of integration was used to include both negative peaks of a typical cortical response[14,31]. Each cortical response with a LL exceeding four

times the connection-specific median LL response was visually inspected to exclude technical artifacts or epileptic discharges.

**Signaling latency.** To account for long-latency cortical responses, we detected responses within 500 ms after stimulation onset as increases in line length over a 250 ms sliding window. We calculated the signaling latency as the latency to the first peak in the average cortical response[33].

**Effective connections.** An effective connection was determined based on the presence of a cortical response in at least some trials. To allow for variable responses and waveforms within effective connections, we derived two potentially contrasted centroid (average) waveforms for each electrode pair, using a K-means clustering of rank two based on Pearson's correlation as a distance metric (Supplementary Fig. 5). An effective connection was deemed significant, when the line length of $\geq$1 cluster centroid was $\geq$95th percentile of the null distribution, drawn from pre-stimulation surrogate data on the same recording electrode ($T_0$ -600 ms to $T_0$)[33]. We grouped effective connections into three categories based on a preset distance (25 mm) and a threshold signaling latency (65 ms) derived from the distribution of signaling latencies within brain regions in the F-tract dataset: (1) local (electrodes A and B within 25 mm in the same hemisphere and brain region), (2) long-range short-latency (>25 mm, $\leq$65 ms), and (iii) long-range long-latency (>25 mm, >65 ms).

**Single-trial cortical responses.** To detect cortical responses on a single-trial level, we evaluated each individual post-stimulation signal for (1) increased magnitude and (2) its waveform resemblance to connection-specific centroids (Supplementary Fig. 5). To do so, we compared a compound metric ($\pm \rho^2 LL$) that multiplies the single-trial signal line length (LL) with its squared cross-correlation coefficient with the centroids ($\pm \rho^2$, max squared Pearson correlation $\pm$10 ms lag with sign preserved) against a null-distribution drawn from non-stimulated baseline data (Supplementary Fig. 5). Intuitively, this approach quantifies the amount of line length attributable to the expected response waveform, as $\rho^2 \to 1$ indicates full resemblance to the expected response waveform, whereas $\rho^2 \leq 0$ suggests cortical activity unrelated to the stimulation.

### Signaling probability and directionality

The signaling probability $P_{AB}$ for a given connection from electrode A to B was calculated as the number of significant responses in B out of total stimulations delivered in A, and inversely for $P_{BA}$. As $P_{BA}$ may or may not equal $P_{AB}$, the corresponding directionality was quantified as follows:

$$DI_{AB} = \frac{P_{AB} - P_{BA}}{\max(P_{AB}, P_{BA})} \qquad (2)$$

where $DI \to 0$ indicates bidirectional signaling ($P_{AB} \simeq P_{BA}$), $DI \to +1$ a pure efference (solely sending) and $DI \to -1$ a pure afference (solely receiving). Note that $DI_{BA}$ is the negative of $DI_{AB}$ and that $|DI| = 0.5$ already means doubling of signaling in one direction, with exponential increases for $|DI| > 0.5$.

### Excitability index

Hourly stimulation-response protocols at 14 stimulation intensities (see above) were aligned to visually labelled sleep-wake stages (below). Measurements with fewer than five trials at a given stimulation intensity and for a given vigilance state were deemed incomplete and excluded. For each electrode pair, we quantified an excitability index (ExI)[14,46] as the area under the stimulation-response curve, plotting

response magnitude (LL of average signal, normalized from 0 to 1 for the maximal LL in wake) against stimulation intensities (normalized as 0% at 0.2 mA to 100% at 12 mA). An ExI→1 reflects maximal sensitivity to minimal stimulation intensity, ExI→0 is non-excitable, and an ExI-0.5 could reflect a linear response increase over the full stimulation range or sensitivity only to higher intensities. An ExI value was considered significant when it exceeded the 95th percentile of pre-stimulation surrogate data.

### Sleep modulation

We used scalp EEG to visually score sleep-wake stages according to the criteria of the American Academy of Sleep Medicine (AASM). Participants 3 and 4 were excluded due to a lack of sufficient sleep data (Supplementary Table 2). Scoring was validated with an average preset Cohen's inter-scorer agreement of >0.70 between two independent scorers (EVM and PN). We investigated the modulation of cortical responses by rapid-eye movements (REM) and non-REM sleep (NREM stages 2-3 combined, stage 1 excluded) compared to wakefulness as the absolute difference in probability and directionality, as well as the percentage change in excitability and response magnitude. For each connection and vigilance state, magnitude was quantified as the line length of the average of ≥5 significant single-trial cortical responses.

### Statistics

The primary measurement in this study was the presence or absence of a cortical response after stimulation of an effective connection at single-trial level, tested against the null distribution of 400 surrogate cortical signals drawn from non-stimulated baseline recordings (Supplementary Fig. 5). We first evaluated resulting signaling probabilities as a distribution across participants regardless of their anatomical location (connection level, Fig. 2). We then grouped and averaged signaling probabilities by single brain region or pair of brain regions at participant level (Figs. 3, 4).

**Connection level.** In our dataset, all connectivity (average response) and signaling (single-trial response) measures were done at the connection level (i.e., for a given electrode-pair) and tested against the 95th percentile of 2 × 200 centroid and 400 single-trial surrogates (one-sided), with and without false discovery rate (FDR) correction for $p < 0.05$, respectively. The significance of the ExI was tested in each connection against the 95th percentile of 200 surrogate datasets with the same number of trials per stimulation intensity. Connection-level data were reported as distributions (P, DI) or averages (ExI) in Figs. 2–4. Differences in directionality by connection type were tested with a Kruskal-Wallis test for the effect of latency and distance (Fig. 2).

**Participant level.** As a complement, to address potential over-representation of some connections in a given participant, we averaged signaling probabilities belonging to a given single or pair of brain region(s) first within (black dots, Fig. 3c) and then across participants (violin, Fig. 3c). A logistic regression model was used to describe the signaling probability as a function of distance (fixed continuous effect), and pairs of brain regions (fixed, interaction effect of stimulation region and response region):

$$logit(P) = \beta_0 + \beta_1 \left( Region_{stim} : Region_{resp} \right) + \beta_2 d \quad (3)$$

Where $P$ is the signaling probability, $logit(P)$ the log-odds, and $\beta_i$ the coefficients. In a post-hoc analysis, their paired difference ($P_{AB}$ vs. $P_{BA}$) was tested with a Wilcoxon signed-rank test (FDR corrected p < 0.05) for limbic (amygdala, hippocampus, parahippocampal and temporo-polar cortex), vs. neocortical categories (all other cortices).

**Sleep modulation.** At the connection level, overall comparison of signaling probabilities across participants and between vigilance states

was done using the Pearson correlation coefficient and a Bland-Altman analysis of bias. At the participant level, changes in signal magnitude and ExI with stimulation in sleep compared to wake were tested with a Wilcoxon signed-rank test (FDR corrected $p < 0.05$) for each pair of brain regions. Changes in signaling probabilities were tested with a mixed-effects linear model for stimulation region (fixed) and participants (random).

**Software.** All data preprocessing, statistical analyses, and visualizations were performed in *Python* (V3.10). Numerical operations and data handling were conducted using *NumPy* (V1.26.4) and *pandas* (V2.2.2). Data visualization was carried out with *matplotlib* (V3.9.1) and *seaborn* (V0.13.2). Statistical testing, including non-parametric comparisons, was performed using *SciPy* (V1.11.1), and multiple-comparison correction (FDR) was applied using *statsmodels* (V0.14.2).

### Reporting summary

Further information on research design is available in the Nature Portfolio Reporting Summary linked to this article.

### Data availability

All measurement data can be explored using the accompanying graphical user interface available at https://github.com/neuro-elab/EvM_Connectivity/releases. In addition, the GitHub repository https://github.com/neuro-elab/EvM_Directed-cortico-limbic-dialogue contains the preprocessed data and analysis scripts required to reproduce the main plots presented in the manuscript. Source data are provided with this paper.

### Code availability

The code for core calculations made in this study are available in the accompanying graphical user interface (see above). In addition, the GitHub repository https://github.com/neuro-elab/EvM_Directed-cortico-limbic-dialogue contains the preprocessed data and analysis scripts required to reproduce the main plots presented in the manuscript.

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

## Acknowledgements

The authors thank Karin Somerlik, Thilo Krueger, and Kornelius Lente at Inomed GmBH for the help with developing a programmable interface with the stimulator and switch matrix. The authors also thank the EEG team at Inselspital for their flexibility in accommodating new devices in the recording chain. We thank the patients who agreed to participate in this study. E. van Maren's salary was supported by the Velux Stiftung (#1232). M.O. Baud's salary was supported by the Swiss National Science Foundation (Ambizione #179929 and Eccellenza #203339). Some electrophysiology material was financed by the Olga Mayenfisch Stiftung, Zürich.

## Author contributions

E.v.M. programmed the recording-stimulation set-up, collected and analyzed the data, generated the figures, and wrote the first draft of the manuscript. C.G.M. provided the final image analysis and electrode localization pipeline and built the interface for data visualization. R.W. analyzed part of the data and built the interface for data visualization. C.F.M. recruited patients and collected the data. P.N. scored sleep data. M.F. built the recording-stimulation set-up and collected the data. J.A. supervised the setup of the recording-stimulation system. C.P. collected the data. A.T. provided scientific input and edited the manuscript. T.P. provided scientific input on Fig. 1 and edited the manuscript. K.S. provided scientific input and edited the manuscript. M.O.B. obtained authorization for the project, built the recording-stimulation set-up, collected and analyzed the data, wrote the first draft and final version of the manuscript, and provided the scientific oversight of the project. All authors edited the final manuscript.

## Competing interests

The authors declare no competing interests.
