## [Transparent Peer Review file · Nature Communications]

Directed cortico-limbic dialogue in the human brain

Corresponding Author: Professor Maxime Baud

Version 0:

Reviewer comments:

Reviewer #1

(Remarks to the Author)

This study aims to map the signal flow in human cortex during wakefulness and sleep using cortico-cortical evoked potentials (CCEPs), testing the null hypothesis of a reversing cortico-limbic dialogue. It is a very interesting study providing innovative results and very detailed analyses, though some of the figures are sometimes difficult to read given the large amount of information.

What are the noteworthy results?

- Directionality index unraveled locally reciprocal connections, but asymmetric signal flow at long-range. Limbic structures send about twice as many signals as they receive, whereas neocortical signaling was largely bidirectional.
- The excitability index, measured in a subset of effective connections over minutes, was found to correlate with signaling probabilities over hours.
- The excitability of limbic afferences and efferences was subtly and bidirectionally modulated during sleep. However, the observed dynamic signaling topology was overall stable, contrary to the prediction of a reversed flow from limbic structures to neocortical regions during sleep in humans.

Strengths:

- Open dataset with strength for being used in both fundamental and clinical future research, particularly for understanding human cortico-limbic interactions;
- Innovative contributions of 1) delivering short-lived electrical pulses over days and 2) assessing the variable fate of each transmitted signal on a single-trial basis instead of average signals;
- CCEPs were recording on both vigilance states: wakefulness and sleep.

Please find below some suggestions to improve the quality of the report:

Major comments:

1) A possible important limitation of the study is that it is not sufficiently clear how the authors move from the novel measure of intertrial variability to measure "signaling probability" to the group map of signaling probability (e.g. fig 1h). The reader interpolates that it is simply an average over all trials, whatever the patient, which gives a mix of intertrial and intersubject variabilities. If it is true, the effect of interindividual variability should be discussed. Also, given the limited patient cohort (n=15), only brain areas sufficiently sampled over the population should be included in the results. Or at least it should be indicated where the population sampling was very limited (e.g. in Fig 3). Also, the authors claim that "our results mark a significant advancement in precisely mapping the millisecond transmission of each delivered probing signal at single-trial resolution, affording the need to capture variable signal flow that was lost in the averages of prior approaches." but they did not present results comparing the two approaches. Including the comparison between traditional averaging vs single-trial resolution approaches regarding cortico-limbic dialogue would enhance the contribution of the study. (discussion)

2) The authors computed maps of signaling probability over hours of wakefulness but did not clearly present and discuss these results. The signaling probability just gives an average measure. How does cortico-limbic dialogue change over hours of wakefulness? E.g. was it correlated with day time? Did the variability had a slow dynamic or was it completely random? A more detailed analysis of cortico-limbic dialogue changes across waking hours would strengthen the contribution of the study. (results: signaling probability)

- 3) The authors should clearly state and discuss the limitations of the study in a dedicated paragraph in the discussion.
- 4) Signaling probability was assessed using a time window up to 600ms. This clearly includes indirect responses and thus makes questionable the rationale on the signaling probability measure. The choice of this long time window should be better justified.
- 5) Another important concern is on the interpretation of the "signaling directionality" measure. As signaling probability is a measure of intertrial variability (maybe not only, see comment 1), it is not fully clear why asymmetry of signaling probability should be interpreted as a directed measure of information flow, thus coined "signaling directionality". It could also be interpreted as a measure of asymmetry of excitability threshold changes to brain states fluctuations. This term is perturbing as in Figure 2a second line, the latency of the responses does contradict the measure of directionality. Figure 2b is difficult to understand. Finally, the values of asymmetry reported (Figure 3a) are surprisingly high, given known neuroanatomy and the fact that the parameters of the "signaling probability" takes into account re-entrant loops. It could be that this map simply reflects regional changes of brain excitability. Please discuss further. Do not understand the difference between second and third line of Figure 3b.

Minor comments:

- 6) The authors hypothesize "a reversing cortico-limbic dialogue" in humans between sleeping and awake states. We understand from the introduction that such reversal in signal flow was already found in animal studies but is unknown in humans. The way the text is phrased is however not perfectly clear as it is mainly composed of strong general statements with numerous references without precise results description. I suggest the authors to identify the specific animals in which this was proved, and what experiments were actually done. As it stands, the rationale for applying the same hypothesis in humans, and why CCEP is a sound approach, does not seem clearly articulated. (introduction)
- 7) The GitHub data link appears broken ("404 This is not the web page you are looking for"). This should be corrected. (data availability)
- 8) Although it is not critical for the study, it is unclear why an old, undersampled thus less accurate, version of the F-TRACT atlas was used for the parcellation study and the definition of direct and indirect connections. If it does not require extensive data reprocessing, it could be interesting to update the Figure 1c with a more appropriate version, unless Louvain's clustering approach gives significantly different results.
- 9) To avoid data over-interpretation, I suggest the authors to use the words "short-latency" instead of "direct" and "long-latency" instead of "delayed" once you are taking into account the signaling latency outcome to define the threshold of 65ms.
- 10) The authors mentioned "signaling probability (P), calculated as the number of significant single-trial cortical responses in electrode B, out of the total number of stimulations delivered in electrode A". I suggest the authors to clearly define how to calculate signaling probability for each "brain community" (results: signaling probability)
- 11) The statement "In a complex network, how often signal is transmitted along a connection (i.e. PAB) may depend on how excitable (i.e. responsive) this connection is." lacks a reference to support it. (results: connection excitability). The same happens for statement "but kept the amygdala and hippocampus separate from the surrounding parahippocampal and temporo-polar cortex, given their unique functions." (methods: cortical parcellation)
- 12) The authors should avoid the use of subjective vocabulary. For example, 1) "intuitively" in line 19 page 6; 2) "high average P" line 1 page 8; 3) "our study's strength is its simplicity" in line 11 page 10.
- 13) The authors referred a waveform response characteristic calculated for single-trials but did not explain how is calculated nor present this result.
- 14) I suggest to isolate the control analyses in a separate figure (supplementary material)
- 15) Signal rescaling method used to normalize amplitudes across recordings should be described in detail to ensure reproducibility. (methods: signals processing).

(Remarks on code availability)

The GitHub data link appears broken ("404 This is not the web page you are looking for"). This should be corrected. (data availability)

Reviewer #2

(Remarks to the Author)

The manuscripts investigated causal interactions between limbic and cortical structures, focusing on single trial level cortico-cortical evoked potential (CCEP) analysis. The authors demonstrate asymmetrical connectivity within cortico-limbic networks and report correlations between the directionality index and excitability index. They also explore state-dependent

changes in connectivity by comparing sleep and wakefulness. These findings provide valuable insights into human informational processing involving the limbic system. However, several concerns and questions regarding the methodology and interpretation of results remain and should be addressed.

Major points

1. LL analysis

It is somehow difficult to follow the methods of Line Length (LL) analysis as detailed in the supplementary material. What is the rationale to adopt the two cluster centers (having two different waveforms in the series of single trials?). The methods of probing cortical response (Suppl Fig 3c) using Pearson's correlation to WOI of CCs and following computation of compound metric is not easily understood for readers from clinical neurology/neurophysiology. The authors are recommended to make this part better understood for these readers.

2. We agree with the authors on the importance of analyzing CCEP at the single-trial level to deepen our understanding of human informational flow and ways of processing. Nevertheless, at the same time, single-trial analysis is inherently susceptible to random noise, and distinguishing physiological variability from stochastic fluctuations remains a methodological challenge. The manuscript benefit from a more detailed explanation of how the authors addressed or controlled for the influence of random noise in their analysis.

3. The authors emphasize the utility of single-trial CCEP analysis in assessing connectivity asymmetry and sleep-related modulation. However, previous studies—such as those by Entz et al. and Usami et al.—have addressed these questions using conventional signal averaging techniques. Averaged waveforms inherently account for trial-level variability by reducing the influence of noise through signal integration. The authors should clarify what unique insights were gained specifically through single-trial analysis that could not have been observed using traditional averaged CCEP approaches. A more explicit discussion on the added value and methodological advantages of their single-trial framework is warranted.

Minor points

1. Centroid Definition and Use in Single-Trial Analysis

The authors applied K-means clustering to derive two centroids from the averaged responses. It remains unclear why two centroids were necessary rather than one representative average response. Additionally, In the Methods section, it is stated that single trials were compared to one of these centroids using correlation coefficients. The authors should clarify which centroid was selected for comparison, and on what basis this selection was made.

2. Definition of Epileptic Electrodes

The manuscript would benefit from a clearer definition of “epileptic electrodes.” Do these refer exclusively to electrodes located in the seizure onset zone (and the seizure propagation zone), or do they also include irritative zones as defined by interictal discharges? Furthermore, how often were interictal discharges present in electrodes labeled as “non-epileptic”?

3. Excitability index: Electrode selection criteria

The authors state that four electrodes were chosen for evaluating ExI, but it is unclear where these electrodes are located. Were all these electrodes located in the hippocampus or did the selection include epileptic electrodes? The inclusion criteria should be mentioned.

(Remarks on code availability)

Reviewer #3

(Remarks to the Author)

(Remarks on code availability)

Reviewer #4

(Remarks to the Author)

I co-reviewed this manuscript with one of the reviewers who provided the listed reports. This is part of the Nature Communications initiative to facilitate training in peer review and to provide appropriate recognition for Early Career Researchers who co-review manuscripts.k

(Remarks on code availability)

Version 1:

Reviewer comments:

Reviewer #1

(Remarks to the Author)

The authors took into consideration reviewers' comments and appropriately revised their work.

(Remarks on code availability)

One can now access the GitHub data link but one cannot see any data or code, only a readme file inside it. This should be checked.

Reviewer #5

(Remarks to the Author)

(Remarks on code availability)

I can now access the GitHub data link (https://github.com/neuro-elab/EvM_Connectivity/releases/tag/Connecto_v0.3) but I cannot see any data or code. I only see a readme file inside it. This should be checked.

We thank the Editor and the four Reviewers for giving us the possibility to revise and strengthen our manuscript. In particular, we thank the Reviewers for recognizing the novelty and the many strengths of our study, while making constructive comments for further improvement of the quality of the report. We thank the Reviewers for recognizing the innovative nature of our approach at single-trial level and the valuable insights into ‘human informational processing’. We are confident that others will build upon our work mapping the fate of individual probed signals in the brain as a method to study brain dynamics that is far more precise than currently available methodology. We agree with the Reviewers that the analyses are detailed and provided rich information. This is what led us to adopt a visually-attractive connectogram for the main figures, and its interactive version as a graphical user interface. We now also provide and expanded supplementary data with additional text and equations to better explain each step of our analyses. We believe that we were able to address the remaining concerns below.

Reviewer #1 (Remarks to the Author)

Major comments:

- 1) A possible important limitation of the study is that it is not sufficiently clear how the authors move from the novel measure of intertrial variability to measure “signaling probability” to the group map of signaling probability (e.g. fig 1h). The reader interpolates that it is simply an average over all trials, whatever the patient, which gives a mix of intertrial and intersubject variabilities.

Reply: We thank the Reviewer for pointing out the need to improve clarity as to what grouping was used to report average results. We agree with the Reviewer, that participants may not have the same signaling probabilities and that connection with consistent signaling probabilities across participants are likely of higher interest.

We defined signaling probabilities as the number of statistically significant responses over the total number of delivered stimulations, measured bi-directionally for a given pair of electrode contacts in the brain. These individual connection results can be explored, using the provided graphical-user interface that can be downloaded here: https://github.com/neuro-elab/EvM_Connectivity/releases/tag/Connecto_v0.3. For a compact presentation of our results, we present them as distributions across participants (Connection-level in Fig. 2-3), and as averages according to two levels sequentially (Participant-level Fig. 3-4):

- 1) Within given single or pairs of brain regions (Connectograms), with brain regions defined in Fig. 1.
- 2) Across participants.

We have further clarified this point in the statistics section: *“The raw primary measurement in this study is was the presence or absence of a cortical response after stimulation of an effective connection at single-trial level, tested against the null distribution of 400 surrogate cortical signals drawn from non-stimulated baseline recordings (Supplementary Fig. 5). signaling probability in one or the other direction of a given pair of electrode contacts stimulated alternately. We first evaluated these resulting signaling probabilities as a distribution across participants regardless of their anatomical location (connection level, Fig. 2). We then grouped and averaged signaling probabilities by single- or pairs of brain regions at participant level (, then across participants (Brain region level, Fig. 3-4).”*

2) If it is true, the effect of interindividual variability should be discussed.

Reply: We want to emphasize that already in our original submission, we had shown the inter-participant consistency of our measurements. First, we reported group averages along with individual averages, for example in figure 3c. In our original Supplementary figure 4f (now supp. Fig. 7c), we also reported the consistency of signaling probabilities in pairs of brain regions observed across participants.

3) Also, given the limited patient cohort (n=15), only brain areas sufficiently sampled over the population should be included in the results. Or at least it should be indicated where the population sampling was very limited (e.g. in Fig 3).

Reply: We want to emphasize, as done in our original discussion, that the strength of our study is not the number of included participants, but the repeated probing of the same connections in those participants, which allows us to discuss actual interregional signaling over time. Already with the presentation of the result of Fig. 1, we acknowledge that our study under-samples the brain, like all icEEG studies: *"Although this topology is undoubtedly under-sampled and biased - iEEG is collected according to clinical priorities - it serves here as a reference 'connectogram' for mapping signal flow across brain regions and vigilance states"*. Since we study changes within established connections over time, this is, in our opinion, a minor issue. This is also what led us to include larger datasets for adopting a brain parcellation for our study.

It is correct that fig. 1h reports the finer-grain average across participants and connections in each pair of cortical area (as opposed to larger brain region). This was done mostly for visual purposes to compare Fig 1h with Fig. 1c. We have highlighted this point in the figure legend. For the other Fig. 3-4, the coarser parcellation of brain regions derived from Fig. 1, we now report the number of tested connections for each region-pair.

4) Also, the authors claim that "our results mark a significant advancement in precisely mapping the millisecond transmission of each delivered probing signal at single-trial resolution, affording the need to capture variable signal flow that was lost in the averages of prior approaches." but they did not present results comparing the two approaches. Including the comparison between traditional averaging vs single-trial resolution approaches regarding cortico-limbic dialogue would enhance the contribution of the study. (discussion)

Reply: We thank the Reviewer for pointing out that we had not yet directly compared the established Z-scored average CCEP approach to our novel signaling probability measure in the calculation of a directionality index (DI). We now include this analysis in in Supplementary Fig 8a and provide the figure below for the Reviewer's convenience.

A DI had originally been proposed by Entz et al, Human Brain Mapping 2014, formulated as the ratio of cortical responses magnitude in one and the other direction. For each bi-directional connection, we have now calculated the DI calculated based on the ratio of magnitudes of the Z-scored average CCEP in one and the other direction (DI_z) and compared it directly to our proposed DI_p based on the ratio of probabilities. We found that the two measurements agreed well when $DI_p \rightarrow \pm 1$, which makes intuitive sense, because if there is no effective connection in one or the other direction, this will be reflected equivalently in the very low response magnitude or probability. However, for DI_p between [-0.9, 0.9], there was a very wide dispersion of DI_z . Of specific concern, many connections in which we found a $DI_p \rightarrow 0$, would have been attributed a $DI_z \neq 0$ (dark vertical high density of points centered on DI_p

= 0). Thus, DI_p and DI_z are measuring different attributes of an effective connection, the former the frequency of signals transmitted in one and the other direction, the later the relative signaling magnitude. For the study of signal flow, we must reject the measurement of directionality by comparing the magnitude of cortical responses, as a reasonable alternative. We thank the Reviewer for suggesting this analysis, which enhanced our contribution. We have now included this analysis in Supplementary Fig 4, along with already-provided correlation analyses.

- 5) The authors computed maps of signaling probability over hours of wakefulness but did not clearly present and discuss these results. The signaling probability just gives an average measure. How does cortico-limbic dialogue change over hours of wakefulness? E.g. was it correlated with day time? Did the variability had a slow dynamic or was it completely random? A more detailed analysis of cortico-limbic dialogue changes across waking hours would strengthen the contribution of the study. (results: signaling probability)

Reply: We thank the Reviewer for this important question. Changes in signaling probability are likely not random, and it is reasonable to hypothesize that they change as a function of the circadian phase or sleep-wake homeostasis. While we feel that an extensive investigation on this question is out of the frame of our study on brain states, we nevertheless provide a global analysis that showed subtle but consistent increases in signaling probability across cortico-limbic connections and participants over the course of the day.

This analysis was done based on wakefulness over one day of data per participant divided into three 5h-blocks (7-12:00, 12-17:00, 17-22:00). Only connections that had been probed ≥ 10 times per block were included. As a non-parametric equivalent to a two-way ANOVA, we used a Schreier-Ray-Hare test for grouping variables amygdala, hippocampal or neocortical connections and daytime block, followed by a post-hoc paired Wilcoxon test across participants.

We found that signaling probabilities changed by connections (already known, $p < 10^{-141}$) and by daytime block (new, $p = 0.001$), without an interaction term ($p = 0.9$), suggesting that connections from and to limbic structures and neocortex change equally over the course of the day.

	Source	H	df	p
0	Connection	653.507616	2	1.237725e-142
1	block	13.783085	2	1.016345e-03
2	Connection:block	0.949385	4	9.173713e-01

In each case, a post-hoc analysis revealed an increase in signaling probabilities over the course of the day, with effect-sizes of about +8-17% across connections.

Connection	N	Effect size		
		morning_afternoon	morning_evening	afternoon_evening
cortex-cortex	8024	+10.1% p<10 ⁻¹⁵	+9.9% p<10 ⁻¹⁵	+1.8% p=0.14
Amygdala-cortex	1099	+8.1% p=0.015	+8.0% p=0.016	+0.8% p=0.82
Hippocampus-cortex	1134	+17.6% p<10 ⁻⁷	+17.0% p<10 ⁻⁶	+0.03% p=0.92

At this stage, we did not include this result into our manuscript. There are certainly myriad other modulations of signaling probabilities, such as relaxing vs. concentrating, the release of neuromodulators (noradrenaline, acetylcholine, etc.), hormonal influences, circadian and homeostatic regulations. Our study seeks to test a long-standing hypothesis of inversed cortico-limbic flow during sleep, compared to wakefulness. Even though we here showed that signaling probabilities may change during wakefulness itself, we believe that the responses provided to our specific study question are still valid. First, we compared signaling probabilities during sleep to signaling probabilities during wake at any hour, averaging out the effects presented here. Second, we showed that sleep influences signaling probabilities in opposite directions in the amygdala and hippocampus, arguing against a systematic bias. Third, we confirmed these results by also comparing the excitability index between sleep and wakefulness.

6) The authors should clearly state and discuss the limitations of the study in a dedicated paragraph in the discussion.

Reply: in our original manuscript, the second paragraph of the discussion is the paragraph dedicated to limitations: *“Nevertheless, some limitations pertain to the collection of sparse and biased iEEG, etc”*. We typically discuss the limitations upfront in the discussion of our results, such that the reader can understand our interpretation and claims in the light of these limitations.

- 7) Signaling probability was assessed using a time window up to 600ms. The choice of this long time window should be better justified.

Reply: We agree with the reviewer that it is important to distinguish direct and indirect effective connections, because the direct responses are more likely to be those linked to structural connectivity (presence of a direct tract). This is why we introduced a data-driven cut-off to distinguish these two types of connections, now termed *short-latency* (<65ms) and *long-latency* (≥ 65 ms). It should be noted that the time-window for line-length calculation is 250ms (to include N1 and N2 of a classical CCEP waveform) within a range of 500ms post-stimulation, not 600ms. The motivation was to not miss any delayed response, accounting for delays in response onset of up to about 200-300ms, similarly to the F-tract dataset, which would then be captured between 250-500ms. Our results show that for all practical purposes, we very rarely found connections with signaling latency >200ms (Sup. Fig. 4). We now revised the explanation on signaling latency in the methods and emphasize this point, by complementing an existing sentence in the results: "... *whereas long-range (≥ 25 mm) signaling across regions could have long-latency (≥ 65 ms, Fig. 1d) 32, typically with a peak-latency of 65-200ms.*"

- 8) This clearly includes indirect responses and thus makes questionable the rationale on the signaling probability measure.

Reply: We respectfully disagree with the Reviewer statements that *the rationale for the signaling probability measure is questionable*, in the context of connections with different latencies. In our study, we made efforts to characterize effective connectivity and signaling within these connections with different attributes, that we then compared to ensure minimal co-linearities (original Supp. Fig. 4, now Sup. Fig. 8). Observing the number of signals transmitted (i.e. the signaling probability) can be done independently of the response delay. Somewhat intuitively, we found that signaling probabilities were lower for long-latency responses (e.g. 65-200ms), but also that they were characterized by their unidirectionality. Such unidirectional long-latency signaling may act as feedback or feedforward modulations in the brain, that likely cannot be captured with structural connectivity alone. There are theoretical reasons to believe that such weak and delayed responses may have large influences on complex networks, and their potential importance will have to be tested in future studies. We now clarified this point in the following comment in the result section: "*Based on this result, we speculate that long-range signaling may help broadcast neural communication at short-latency and create directional feedforward or feedback loops at long-latency*".

- 9) Another important concern is on the interpretation of the "signaling directionality" measure. As signaling probability is a measure of intertrial variability (maybe not only, see comment 1), it is not fully clear why asymmetry of signaling probability should be interpreted as a directed measure of information flow, thus coined "signaling directionality".

Reply: We want to highlight that we never used the term "information flow", but always adhered to the term "signal flow", indeed we do not know if the measured signals carry information in our study. Many large-scale connectivity studies relying on fMRI or EEG have made bold claims in the past about measurements of signaling directionality and/or information flow. On one side, fMRI relies on BOLD signal, which is not a neural signal,

but a neurovascular one. On the other side, fMRI and EEG cannot measure directionality causally, as they do not probe the brain. In contrast, our measurement of signal directionality is based simply on the number of probed signals observed in one and the other direction for a given connection. We want to emphasize again that all measurements (probability and DI) are done first at the connection-level and single-trial level, as stated and now clarified in our statistics section in the methods, asking the question: ‘Given an effective connection, how often (probability) does it signal in one and the other direction’. We have also slightly reformulated parts of the section signaling probability and directionality in the result section to further clarify this point. For example, if we stimulate electrode A (B) 100x and observe 90 (30) responses at recording electrode B (A), we would conclude that chances of signaling from A to B (90/100) are 3x those of signaling from B to A (30/100). Based on this intuition, we propose a definition of signaling directionality as a ratio of observed probabilities, or in other words, a ratio of the number of transmitted signals in either direction. Thus, our proposed measurement of signaling directionality at the connection level is not influenced by the incidence of effective connections (i.e. the likelihood of finding an effective connection between two brain regions across patients).

10) It could also be interpreted as a measure of asymmetry of excitability threshold changes to brain states fluctuations.

The proposed measurement of signaling probability is purely observational and does not rely on any assumption regarding an underlying mechanism. It could indeed relate to differences in the excitability of the connection in one or the other direction, as the Reviewer suggests and as we discuss in the results of the original Figure 2e. We agree that it is important to take brain and vigilance states into account, which is what we do in the original Figure 4.

11) This term is perturbing as in Figure 2a second line, the latency of the responses does contradict the measure of directionality.

Reply: We did not report the response latencies in Fig. 2a and are unsure of what the Reviewer was referring to. However, we agree with the Reviewer that, in theory, the response latency in one and the other direction might relate to the directionality of a given effective connection. If that was the case, measuring the latency in one and the other direction would be shorter as a proxy of directionality than probing numerous times to compute the probability of signal transmission. As a matter of fact, we had asked ourselves the question and had done a supplementary analysis in our original Supp. Fig. 4 (now Suppl Fig. 8) comparing the delta peak latency and our DI. We here provide this analysis for the Reviewer’s convenience. It shows the absence of linear relationship between the delta peak latency and our DI.

12) Figure 2b is difficult to understand.

Reply: Fig. 2b is simply the distribution (probability density function) for the two key metrics of the article (signaling probability and directionality) in three types of connections, classified by delay and distance. We have rephrased the panel legend and believe that this helps to clarify what is being displayed.

13) Finally, the values of asymmetry reported (Figure 3a) are surprisingly high, given known neuroanatomy and the fact that the parameters of the “signaling probability” takes into account re-entrant loops. It could be that this map simply reflects regional changes of brain excitability. Please discuss further.

Reply: We thank the Reviewer for pointing this out. To enhance contrast between regions, we had decided to plot the regional summary in Fig.3a against a different scale than Fig.3b, as indicated in the original legend: “*Green regions are biased towards sending, purple towards receiving (not at scale with legend)*”. The Reviewer’s comment shows that we may have created more confusion than clarity with this choice. In the revised version, we simply plotted Fig. 3a on the same scale as Fig. 3b, and the new panels are provided below for the Reviewer’s convenience. Thus, directionality can be high in a given connection (Fig.3b), but average regional biases are low (Fig. 3a). Regional differences in excitability can also be found in the original Supplementary Fig. 5d (now Supp. Fig. 9).

14) Do not understand the difference between second and third line of Figure 3b.

Reply: The top row (second line) are the connection of neocortex with limbic structures, whereas the bottom row (third line) are the connections among neocortical areas, limbic regions excluded. We have clarified this further in the figure legend.

Minor comments:

6) The authors hypothesize "a reversing cortico-limbic dialogue" in humans between sleeping and awake states. We understand from the introduction that such reversal in signal flow was already found in animal studies but is unknown in humans. The way the text is phrased is however not perfectly clear as it is mainly composed of strong general statements with numerous references without precise results description. I suggest the authors to identify the specific animals in which this was proved, and what experiments were actually done. As it stands, the rationale for applying the same hypothesis in humans, and why CCEP is a sound approach, does not seem clearly articulated. (introduction)

Reply: We thank the Reviewer for helping us improve our introduction and the rationale for our hypothesis. We have made extensive changes to the introduction in that sense, including:

- Inverting the rodent and human paragraphs for improved flow and highlighting the limitations in human as compared to rodent research.
- Introducing the historical discovery of the circuit of Papez early, emphasizing that while the structure is known, its dynamics are not.
- Specifying that the hypothesis for a reversing cortico-limbic dialogue mainly stems from rodent work.
- Re-emphasizing that icEEG is superior to fMRI and EEG in its ability to directly measure brain signals.

7) The GitHub data link appears broken ("404 This is not the web page you are looking for"). This should be corrected. (data availability)

Reply: The following link should work. We apologize for the error the previous link had generated, as the link was still in private and not public mode.

https://github.com/neuro-elab/EvM_Connectivity/releases/tag/Connecto_v0.3

8) Although it is not critical for the study, it is unclear why an old, undersampled thus less accurate, version of the F-TRACT atlas was used for the parcellation study and the definition of direct and indirect connections. If it does not require extensive data reprocessing, it could be interesting to update the Figure 1c with a more appropriate version, unless Louvain's clustering approach gives significantly different results.

Reply: We used the P10_v2210 F-tract dataset in our initial submission and ran the Louvain algorithm on that dataset. We thank the Reviewer for pointing out this inconsistency and updated the legend of Figure 1 in that sense.

9) To avoid data over-interpretation, I suggest the authors to use the words "short-latency" instead of "direct" and "long-latency" instead of "delayed" once you are taking into account the signaling latency outcome to define the threshold of 65ms.

Reply: we thank the Reviewer for being careful with terminology and avoid interpretative terms. We adjusted the terminology accordingly.

10) The authors mentioned "signaling probability (P), calculated as the number of significant single-trial cortical responses in electrode B, out of the total number of stimulations delivered in electrode A". I suggest the authors to clearly define how to calculate signaling probability for each "brain community" (results: signaling probability)

Reply: In our original submission, we provided two levels of results: connection and participant level. At *participant* level, the average signaling probability reported per pair of cortical area or brain region is simply the average across connection found in a given patient for a given region-pair, and then the average across participants, such that each participant contributes equally to the grand average.

For improved clarity, we have now further specified that in the methods: *"as a complement, to address potential overrepresentation of some connections in a given participant, we averaged connections belonging to a given single or pair of brain region(s) first within and then across participants."*

Our original explanation in the Fig. 1h legend was indeed not complete enough 'h: Map of signaling probability averaged across 15 participants over hours of wakefulness' and we have now revised to 'h: Map of average signaling probability averaged within a pair of cortical areas and then across 15 participants over hours of wakefulness.'

11A) The statement "In a complex network, how often signal is transmitted along a connection (i.e. PAB) may depend on how excitable (i.e. responsive) this connection is." lacks a reference to support it. (results: connection excitability).

Reply: we believe that this is an intuitive hypothesis, based on our own reasoning. Hence, the phrasing 'may depend on' which emphasize its possibility. We now open the next sentence with 'To test this...' to again emphasize that this is a hypothesis tested in Fig. 2. We do not know a specific reference for this hypothesis.

11B) The same happens for statement "but kept the amygdala and hippocampus separate from the surrounding parahippocampal and temporo-polar cortex, given their unique functions." (methods: cortical parcellation).

Reply: We now re-cited references that were already part of the introduction for this specific statement in the methods section.

12) The authors should avoid the use of subjective vocabulary. For example, 1) "intuitively" in line 19 page 6; 2) "high average P" line 1 page 8; 3) "our study's strength is its simplicity" in line 11 page 10.

Reply: We agree with the Reviewer that scientific text should avoid subjective vocabulary and we have revised the statements as requested (P6L11 and P8L1). However, to help the reader understand a study, the discussion must state its strengths as well as its limitations. We do not find the term 'simplicity' particularly subjective when it comes to computation. Simplicity can be viewed as the inverse of the number of parameters used in fitting models. Here we did not deploy any deep- or machine-learning and therefore we consider our calculations simple. We kindly request to keep this sentence on the strengths of our study.

13) The authors referred a waveform response characteristic calculated for single-trials but did not explain how is calculated nor present this result.

Reply: in our original manuscript, we mainly refer to the CCEP waveform in the methods section of the main text. In addition to the examples of CCEPs provided in Fig. 1, 2 and 4, we also show other waveforms in the supplementary materials and explain how the waveform is taken into account in the detection of cortical responses. To not complexify things and since original Supplementary Figures 1-3 (now Supp. Fig. 2-5) are already cited in the result section,

we did not make additional references to the CCEP waveform, which was a minor methodological point of our study.

14) I suggest to isolate the control analyses in a separate figure (supplementary material)

Reply: it is not clear to us to which control analyses the reviewer refers? Analyses controlling for laterality and belonging to the epileptic network were already in the Supplementary data in the original submission (original Supp. Fig. 5, now Supp. Fig. 10).

15) Signal rescaling method used to normalize amplitudes across recordings should be described in detail to ensure reproducibility. (methods: signals processing).

Reply: we now provide one additional sentence in the methods on this pre-processing step.

The GitHub data link appears broken ("404 This is not the web page you are looking for"). This should be corrected. (data availability)

Reply: as above.

Reviewer 2 and Co-reviewers 3-4

Major points

1. LL analysis. It is somehow difficult to follow the methods of Line Length (LL) analysis as detailed in the supplementary material. What is the rationale to adopt the two cluster centers (having two different waveforms in the series of single trials?).

Reply: We have now largely extended the supplementary data with explanatory text to clarify our methods. The rationale to adopt two cluster centers is to accommodate variability in the presence, absence and specific waveform of the cortical responses as detailed now in the supplementary data.

The methods of probing cortical response (Suppl Fig 3c) using Pearson's correlation to WOI of CCs and following computation of compound metric is not easily understood for readers from clinical neurology/neurophysiology. The authors are recommended to make this part better understood for these readers.

Reply: as above, the additional explanatory text should clarify our methods. We have added an intuitive statement about the compound metric representing 'the line-length attributable to the expected cortical response'.

2. We agree with the authors on the importance of analyzing CCEP at the single-trial level to deepen our understanding of human informational flow and ways of processing. Nevertheless, at the same time, single-trial analysis is inherently susceptible to random noise, and distinguishing physiological variability from stochastic fluctuations remains a methodological challenge. The manuscript benefit

from a more detailed explanation of how the authors addressed or controlled for the influence of random noise in their analysis.

Reply: We agree with the Reviewer that our key methodological improvement must be made clear throughout our manuscript. We thank the Reviewer for pointing out the lack of clarity on our statistical methods in the original manuscript.

Essentially, we adopted a surrogate timeseries approach in which we compute the values expected from the inherent iEEG physiological and random variability ('noise') for our different statistics (e.g. line length, compound metric, etc.). In each case, we draw a null distribution of these values from non-stimulated iEEG data (baseline recordings) and test whether our post-stimulation values surpass ($\geq 95^{\text{th}}$ percentile) that null distribution.

To improve clarity of the main text, we added the following phrase to the result section (P3L40) to make our statistical strategy clearer: "For each single trial, the significance of a cortical response was tested against the expected signal variability found in non-stimulated baseline recordings".

In our main methods, we specified: "Each hourly block of cortical mapping entailed a 5' baseline non-stimulated period, followed by 3-5 stimulations per electrode (electrode 'A') while recording from all others (electrode 'B', 'C', etc.)." And in our statistical section, we again specify: "The primary measurement in this study was the presence or absence of a cortical response after stimulation of an effective connection at single-trial level, tested against the null distribution of 400 surrogate cortical signals drawn from non-stimulated baseline recordings (Supplementary Fig. 5)."

In our supplementary data, we now explain our statistical methods to control for expected variability in depth (Supplementary Fig. 5).

3. The authors emphasize the utility of single-trial CCEP analysis in assessing connectivity asymmetry and sleep-related modulation. However, previous studies—such as those by Entz et al. and Usami et al.—have addressed these questions using conventional signal averaging techniques.

Reply: we agree that Entz et al., Human Brain mapping, 2014 and Usami et al., Human Brain mapping, 2015 are two important prior work, which we had included as references in our original submission. We discuss specific comparisons to these studies below.

Averaged waveforms inherently account for trial-level variability by reducing the influence of noise through signal integration.

Reply: signal averaging yields the first statistical moment of a cortical response by extracting its central tendency. This is the way CCEPs have traditionally been calculated and indeed cancels out stimulation-unrelated signal fluctuations. The variability of cortical responses around this central tendency can be captured by the variance (or standard deviation) which

represents the second statistical moment of CCEPs (e.g. Zelman et al., *Neuron*, 2023.). Given that cortical responses to intracortical stimulation are strong signals (signal-noise ratio $\gg 1$), we opted to evaluate them at single-trial level, while accounting for physiological fluctuations in signal (see compound metric in methods).

The authors should clarify what unique insights were gained specifically through single-trial analysis that could not have been observed using traditional averaged CCEP approaches. A more explicit discussion on the added value and methodological advantages of their single-trial framework is warranted.

Reply: In essence, the advantage of our single-trial framework is the study of signal flow within a network of effective connections. Thus, our method goes beyond other CCEP studies, in that it did not stop at studying effective connectivity but assessed time-varying signal flow. We have now specified this point in the result section: “Thus, our single-trial framework goes beyond the assessment of effective connectivity, as it affords the need to capture time-varying signal flow.” We have also expanded the discussion of this key strength of our study in the discussion: “Beyond the assessment of effective connectivity, our single-trial framework marks a significant advancement in precisely mapping the millisecond transmission of each delivered probing signal, affording the need to capture momentary signal flow that was lost in the averages of prior approaches.”

We here provide a more specific answer to the points raised:

- 1) Directionality: Entz et al. defined a directionality index (DI) based on the ratio of the CCEP amplitude in one and the other direction. In their calculation a $DI > 1$ means outgoing (efferent), whereas a $DI < 1$ means incoming (afferent). They observed many connections with $> 50\%$ difference in amplitude in either direction, which they deemed directional connections. In their figure 3, one can see a continuum of resulting ratios. While this is an interesting result, a ratio of response amplitude may reflect variations in connection strength but does not reflect how signaling varies over time. Entz DI answers the question ‘*how strong* is the CCEP response in one and the other direction’. In contrast, we defined a directionality index based on the number of cortical responses observed over time in one or the other direction among the total delivered stimulations. Our DI answers the question ‘*how often* is signal transmitted in one and the other direction’, over hours of recording. It is a measure of signaling dynamics within an effective connection, rather than a measure of response strength. We have now also included a direct comparison between Entz method and ours in supplementary Fig. 8a, in response to a similar point raised by Reviewer 1. Overall, our study goes beyond that of Entz et al. by providing the actual direction of signaling over time.
- 2) Sleep modulation: Usami et al. investigated changes in mean CCEP amplitude and High-gamma activity according to vigilance stages among 45 recording sites. They found that the CCEP amplitude and High-gamma activity tended to increase in NREM sleep, while it decreased back to wake levels during REM sleep. In contrast, we

evaluated the magnitude, excitability (Fig. 4) and the probability (Supp. Fig. 13) of cortical responses during NREM and REM sleep in a brain-region specific approach across 37,850 effective connections. Our results concur with those of Usami, in that the modulation of cortical responses by NREM were usually stronger than by REM. However, the grouping of connections by pairs of brain regions was key in our case. For example, connections from the amygdala and hippocampus to the neocortex were modulated in opposite directions by sleep (Fig. 4). Overall, our study goes beyond that of Usami et al. by providing sleep modulations of signaling dynamics for specific brain regions.

Minor points

1. Centroid Definition and Use in Single-Trial Analysis

The authors applied K-means clustering to derive two centroids from the averaged responses. It remains unclear why two centroids were necessary rather than one representative average response. Additionally, In the Methods section, it is stated that single trials were compared to one of these centroids using correlation coefficients. The authors should clarify which centroid was selected for comparison, and on what basis this selection was made.

Reply: we derived two centroids from the individual cortical responses, not the averaged response. These represent two potentially contrasted averages across two clusters of responses, instead of only one across all responses, affording the need to capture cortical of variable waveform in the same effective connection. We did not choose one centroid to compared individual responses a priori. We compared each individual response with both centroids and retained the maximal cross-correlation. We understand that the use of the singular in the method section may have led to confusion and now rephrased in the plural form: “their waveform resemblance to connection-specific centroids (Supplementary Fig. 5)”. As stated above, the additional explanatory text in the supplementary data should now clarify our methods.

2. Definition of Epileptic Electrodes

The manuscript would benefit from a clearer definition of “epileptic electrodes.” Do these refer exclusively to electrodes located in the seizure onset zone (and the seizure propagation zone), or do they also include irritative zones as defined by interictal discharges? Furthermore, how often were interictal discharges present in electrodes labeled as “non-epileptic”?

Reply: as stated above, we have now expanded our supplementary data to provide additional explanation on our methodology. In our original submission we had included a control analysis to compare the afferent and efferent signaling probability between epileptic and non-epileptic hippocampus/amygdala. Based on the clinical workup, each electrode was labeled based on its involvement as either SOZ (part of seizure onset zone), IED (have shown inter-ictal epileptic discharges), propagation (part of seizure propagation zone), or uninvolved. In the legend, non-epileptic

parenchyma was specified as: not involved in ictal discharges, occasional interictal discharges acceptable) and amygdala (not involved in ictal nor interictal discharges). This different criterion for the hippocampus is justified by the fact that even in unilateral mesio-temporal lobe epilepsy, the contralateral hippocampus almost always generates occasional spikes. In our experience, this can be one spike every 15 minutes or so.

3. Excitability index: Electrode selection criteria
The authors state that four electrodes were chosen for evaluating ExI, but it is unclear where these electrodes are located. Were all these electrodes located in the hippocampus or did the selection include epileptic electrodes? The inclusion criteria should be mentioned.

Reply: Given the focus of our study, two electrodes were chosen in the healthy or epileptic hippocampus and amygdala in priority in each participant, and additional two electrodes were chosen randomly in neocortex. We now mention these inclusion criteria in the supplementary data.